# Mechanical and Dynamic Mechanical Properties of the Amino Silicone Oil Emulsion Modified Ramie Fiber Reinforced Composites

**DOI:** 10.3390/polym13234083

**Published:** 2021-11-24

**Authors:** Liping He, Fan Xia, Yuan Wang, Jianmin Yuan, Dachuan Chen, Junchao Zheng

**Affiliations:** 1State Key Laboratory of Advanced Design and Manufacturing for Vehicle Body, College of Mechanical and Vehicle Engineering, Hunan University, Changsha 410082, China; bekate@163.com; 2Department of Vehicle Body Testing Research, CATARC Automotive Test Center (Tianjin) Co., Ltd., Tianjin 300300, China; wangyuan2021688@163.com; 3College of Materials Science and Engineering, Hunan University, Changsha 410082, China; pangyuan2916@hnu.edu.cn; 4College of Civil Engineering, Hunan University, Changsha 410082, China; 13707311929@163.com; 5School of Metallurgy and Environment, Central South University, Changsha 410083, China

**Keywords:** polymer-matrix composites, natural fiber, mechanical properties, dynamic mechanical analysis, interface

## Abstract

The mechanical and dynamic mechanical properties, interface adhesion and microstructures of the amino silicone oil emulsion (ASO) modified short ramie fiber reinforced polypropylene composites (RFPCs) with different fiber fractions were investigated. The RFPCs were made through a combined process of extrusion and injection molding. Mechanical property tests of the RFPCs revealed enhancements in tensile and flexural strengths with increase of the fiber fraction due to the high stiffness of the fiber filler and a better interfacial bonding from ASO treatment. The dynamic mechanical analysis (DMA) results indicated that fiber incorporation plays an important role in DMA parameters (storage modulus, loss modulus, and damping ratio) at *T_g_* by forming an improved interfacial adhesion and providing more effective stress transfer rate and energy dissipation between matrix and fiber. The phase behavior analysis suggests all the RFPCs are a kind of heterogeneity system based on the Cole-Cole plot analysis.

## 1. Introduction

Natural fibers are considered as one of the eco-friendly materials, which have good properties compared to synthetic fibers and achieve an increasing attention because of their environmental friendliness, non-toxicity, sustainability, light weight, lower energy requirements, cost-effectiveness, abundant availability, biodegradability, and so on [1,2]. Synthetic fiber reinforced composites (SFPCs) are replaced by natural fiber reinforced composites (NFPCs) in fields like building construction, furnishings, automobile, and aeronautics engineering, especially in automobile interior [3,4]. Renewable and sustainable natural fiber as an alternative for petroleum-based materials prevents global warming by reducing generating carbon dioxide, which would be a solution to decline of petroleum supplies [5,6]. Thus, the use of NFPCs will satisfy a greener, more sustainable development demand compared to SFPCs [7].

Since NFPCs first appeared, most researchers have focused on the static mechanical properties of materials [8,9,10,11]. Many reports found that mechanical properties of NFPCs are effected by compositions and physical properties like length diameter ration of natural fiber, fiber content, and manufacturing techniques [12,13,14,15,16]. However, as the NFPCs went through various types of dynamic stress during their lifespan, studying on their viscoelastic properties reflects the performance under actual conditions of usage [17]. Damping is one of important parameters for viscoelastic composite materials. The vibration causes undesirable noise and material fatigue, which shortens the lifespan of integral structure. Therefore, it is an appealing challenge to make use of damping behavior of NFPCs in vehicle interior, sports equipment like skateboard, engine cover and so on. The damping behaviors of the NFPCs are even more intricate compared with the SFPCs [18,19,20]. Because natural fibers used as the reinforcements are different from conventional artificially synthesized fibers. Natural fibers are made up of pectin, hemicellulose, lignin(amorphous) and cellulose [21]. This structure have the intrinsic ability to translate vibration to heat by dissipating energy during mechanical process. Hence, the behavior of the natural fibers must be considered as viscoelastic [22,23]. Additionally, the irregular surface and hydrophilicity of the natural fibers makes microstructure at the interface of the NFPC is more complicated than the conventional SFPC [24]. Moreover, NFPCs exhibit more complex energy dissipation mechanisms due to more intricate interface [25,26]. Therefore, the viscoelastic properties of the NFPC are not only depending on the properties of fiber and matrix, but also influenced by the quality of interface.

One of effective ways to evaluate viscoelastic properties and interfacial features of the fiber reinforced polymer system is dynamic mechanical analysis (DMA). The storage modulus (*E*′), loss modulus (*E*″), and damping factor (*tan δ*) obtained from DMA are temperature and frequency dependent, which provide information about interfacial bonding of composites [27,28]. The storage modulus (*E*′) is associated with stiffness and rigidity of the composite material, while the loss modulus (*E*″) and loss factor (*tan δ*) represent the viscous response of the materials and are associated with the loss of internal energy caused by plastic deformation, internal friction, relative molecular motion, relaxation processes, phase transitions, and morphological changes [27]. The dynamic mechanical properties provide molecular-level information to understand the mechanical properties of materials [28]. They also provide information of transition temperature (*T_g_*). T. Khuntia et al. [29] used DMA to study viscoelastic properties of coir fibers reinforced polypropylene composites. DMA results indicated an enhanced storage modulus value and lower damping properties for the composites than the neat PP, which means a better interface bonding and higher stiffness of the composites. M. Mittal et al. [30] studied the viscoelastic behavior of natural fiber reinforced composites with varying fiber content. The results showed that *E*′, *E*″, and *T_g_* of the composite increase with the incorporation of natural fiber. M. Asim et al. [31] investigated the dynamic properties of hybridized natural fiber reinforced phenolic composites through DMA and found that properties of interface affect the dynamic mechanical properties of composites.

Among various natural fibers, ramie fiber (RF) has relatively high specific strength. and good thermal stability [32]. Thus, RF is a good substitute for synthetic fiber as the reinforcement in composites. Polypropylene (PP) resin is most widely used in automobiles due to its corrosion resistance, recyclable, thermal stability, waterproof, and low-cost [33]. Therefore, ramie fiber reinforced polypropylene composites (RFPCs) are environmentally friendly and ideal substitutes for SFPCs. However, RF and PP have different chemical polarity, which leads a poor compatibility and interfacial adhesion between RF and PP. Therefore, an effective method is necessary to obtain a strong bonding between matrix/fiber for an effective transfer of stress and desirable mechanical performances of RFPCs. Amino silicone oil (ASO) is a widely used natural fiber surface treatment agent. It can not only make natural fiber soft and smooth, but also reduce the hydrophilicity of natural fiber surface [34]. Studies by our group have shown that the polar amino groups in the ASO molecules can interact with the hydroxyl groups of the natural fiber and hydrophobic amino silicone oil film are formed on the fiber surface, thereby making the significantly improved various properties of the natural fiber [35]. R. Sepe et al. [36] have studied effects of different chemical treatments on mechanical properties of NFPCs. Results indicated that silane treated fiber composites had better mechanical properties than alkali treatment.

ASO is a kind of modification method for plant fiber. However, nothing has reported dynamic mechanical properties of this new ASO modified RFPC. In present work, DMA was used to understand the mechanical properties of materials based on molecular-level information. RFPCs were fabricated with ASO emulsion to change the surface features of RFs. The influences of ASO modification and fiber addition on the mechanical and dynamic mechanical properties of the RFPCs were studied by varying the fiber fractions (5 wt.%, 10 wt.%, 15 wt.%, 20 wt.%, 25 wt.% and 30 wt.%). Effects of ASO modification and fiber addition on the bonding quality at the interface were estimated by DMA results and SEM. This work also made efforts to understand the phase behavior and structural change of the RFPCs with Cole-Cole plot. It is expected to understand more about static and dynamic mechanical properties of RFPCs will benefit their widely applications both in vehicle and civil engineering.

## 2. Experimental

### 2.1. Raw Materials

RF was from the Institute of Fiber Crops, Chinese Academy of Agricultural Sciences. Polypropylene (PP1100) is from Lanzhou Petrochemical Company, Lanzhou, China. Amino silicone oil (chemical pure) was supplied by Zhuangjie Institute of Chemical Industry, Guangzhou, China. The average molecular weight of Amino silicone oil was approximately 15,000 and the amino value was about 0.5–0.6. The specific material parameters of RF and PP are shown in Table 1 and Table 2, respectively.

### 2.2. Surface Modification

RF with 3–5 mm length was ASO pre-treated before fabricating RFPCs in following process through the modified method invented by He et al. [35]. First, the RFs were ultrasonically dispersed and soaked in the ASO emulsion solution for 3 h at 50 °C. Then wet fibers were put into the vacuum drying oven (Shanghai Binglin Elecctronic technology Co. Ltd., Shanghai, China) for 12 h at 80 °C to remove the moisture.

### 2.3. Fabrication of the RFPCs

A double twin-screw extruder TE-35 (Keya Chemical Equipment Company, Jiangsu, China) is applied to mix the polypropylene (PP) with different amount of the modified ramie fiber (5 wt.%, 10 wt.%, 15 wt.%, 20 wt.%, 25 wt.% and 30 wt.%). The mixed temperature was set in range of 170 °C–190 °C. The mixture was then chopped into pellets. The pellets were then used to produce the standard specimens of the RFPCs with different fiber contents by the HDX50 injection machine (Huada Plastic Machinery Co., Hangzhou, China) under the pressure of 80 MPa at 190 °C. The RFPCs with the fiber incorporation of 5 wt.%, 10 wt.%, 15 wt.%, 20 wt.%, 25 wt.%, and 30 wt.% are labeled as 5RFPC, 10RFPC, 15RFPC, 20RFPC, 25RFPC, and 30RFPC, respectively.

### 2.4. Characterization and Testing

#### 2.4.1. Fourier Transform Infrared Spectroscopy (FTIR) Characterization

The RFs before and after ASO treatment were characterized with FTIR. FTIR spectra of RFs was obtained in Nicolet 5700 FTIR spectrophotometer (Thermoelectric, Madison, WI, USA) with KBr pellet method method. The resolution is 4 cm^−1^ in the region of 500–4000 cm^−1^.

#### 2.4.2. SEM and EDX Characterization

The scanning electron microscopy (SEM) images of RFs and fracture surfaces of RFPC were scanned by the SEM equipment Hitachi S-4800 to study the differences before and after ASO treatment. The samples were pre-treated in vacuum by sputtering gold before scanning. The elemental detection of RFPCs before and after ASO modification via electron dot-mapping was conducted by energy dispersive X-ray microanalysis (EDX).

#### 2.4.3. Mechanical Testing

The tensile data of the unmodified and modified RFPCs was obtained according to the test criteria ASTM D638-10, whereas the flexural data was obtained according to the test criteria ASTM D790-10. The mechanical test equipment is Instron 5985. The flexural test was using the three-point bending method. Impact test machine (CBL-11J) was used to get the impact data of the unmodified and modified RFPCs according to the test criteria ASTM: D256-10. All the results were obtained from the average value of five tests.

#### 2.4.4. Dynamic Mechanical Testing

The RFPCs with different fiber content were analyzed through DMA to estimate the effects of RFs on the dynamic mechanical properties of RFPCs. The dynamic mechanical analysis meter (DMA 242 E Artemis, NETZSCH Scientific Instruments Trading (Shanghai)Ltd., Shanghai, China) was used to obtain storage modulus (*E*′), loss modulus (*E*″), and damping factor (*tan δ*). Cole-Cole plot was painted from obtained storage modulus (*E*′) and loss modulus (*E*″). The dimension of RFPC specimens in DMA test was 5 × 10 × 3 mm^3^. Three-point bending was conducted under a fixed frequency of 1 Hz. The temperature range was −60 °C to 150 °C with a heating rate of 5 °C/min.

## 3. Results and Discussion

### 3.1. Characterization of RFs

SEM images of the RF show a smooth and clean surface of RF before treatment in Figure 1a. It is apparent that the surface of RF is rougher after ASO treatment in Figure 1b. ASO treatment increase the surface roughness of RF, resulting in increment of the specific surface area of RF. It indicates that ASO was coated on the surface of RFs, forming a rough surface with micro nano structure to improve the interfacial compatibility and bonding with resin.

FTIR spectra of RFs before and after ASO modification are shown in Figure 2. It is obvious that after ASO treatment, the two absorption peaks appearing at round 791 cm^−1^ and 1225 cm^−1^ are attributed to flexural vibration of Si-C bonds and symmetrical deformation of Si-CH_3_, respectively [36]. The absorption peaks around 3350 cm^−1^ and 2830 cm^−1^ are the stretch vibration of O-H and the stretch vibration of C-H in both spectra of unmodified and modified fibers, respectively [32]. The decrease in peaks of O-H stretch vibration indicates that ASO coated the active hydroxyl groups on the RFs. According to the results, it is concluded that RFs are successfully modified after ASO treatment and ASO coated the hydroxyl groups on the fiber and increase hydrophobicity of fibers.

### 3.2. Characterization of Unmodified and ASO Modified RFPCs

One of important parameters influencing the properties of the RFPCs is the bonding quality at interface. To investigate the interaction at the fiber-matrix interface, fractured surface of the RFPCs through impact tests in typical SEM images are shown in Figure 3. Holes and evident gaps between matrix and the untreated fiber appear, which suggests the fiber/matrix debonding at interface in the untreated RFPCs as shown in Figure 3a,c. In contrast, no evident holes and gaps between matrix and treated fibers exists in ASO treated RFPCs as shown in Figure 3b,d. Fiber breakages were seen also seen from Figure 3d.

Table 3 shows the surface elemental analysis of RF illustrated in Figure 3c,d through EDX. Carbon is dominant of all the detective elements. The content of carbon in point 1 in untreated RFPC is lower than that in point 2 of ASO treated RFPC due to the adhesion of PP matrix which can be seen in Figure 3d, because the carbon content of PP is higher than that of fibers. Another difference obtained is the presence of silicon in point 2 and point 3 of ASO treated RFPC caused by ASO treatment procedure. The O/C ratio of modified RFPC is lower than unmodified RFPC, which means the lower polarity and better hydrophobicity of RF in the modified RFPC. It is also obtained that PP matrix adhesion on RF is higher in modified RFPC than unmodified RFPC, which is consistent with the images from SEM. Based on the results from SEM and EDX, the ASO modification helps to form a better interface bonding at interface. RFs are distributed in the matrix, when the composite receives external forces, the stress of the matrix transfers to the fiber. RF carries most of the stress in the composite, so a good interface bonding means an effective stress transfer between fiber and matrix.

### 3.3. Mechanical Properties of RFPCs

#### 3.3.1. Physical Properties of RFPCs

Table 1 presents the physical properties of neat PP and modified RFPCs. Experimental densities of neat PP and modified RFPCs were calculated according to ASTM 2734-70 using the electronic densometer with resolution of 0.001. Theoretical densities of modified RFPCs were calculated according to the simple rule of mixture:(1)$$\rho_{c}=\rho_{f}V_{f}+\rho_{m}\left(1-V_{f}\right)$$
where $\rho_{c}$, $\rho_{f}$ and $\rho_{m}$ are densities of the composite, fiber and matrix, respectively. $V_{f}$ is the volume fraction of the fiber.

The porosity of the composite was calculated by Equation (2):(2)$$P=100\%\cdot \left(\rho_{c}-\rho_{e}\right)/\rho_{c}$$
where $\rho_{c}$ and $\rho_{e}$ are densities of the theoretical and experimental values of the composite, respectively. P is the porosity of the composite.

Values of the average particle distance were calculated according to the following Equation (3) [37]:
(3)$$d=\left(\sqrt[{3}]{\left(4\pi \right)/\left(3V_{f}\right)}-2\right)r$$
where d is the average distance between particles. r is the average radius of fibers.

The values of interface surface area for the composites were calculated according to the Equation (4) found by Nelson and Hu [37]:
(4)$$S_{i}=\left(3V_{f}\right)/r$$
where $S_{i}$ is the interface surface area of the composite.

It can be seen in Table 4, as fiber content increased, the porosity and interfacial area of the composite increased. The highest porosity content of 4.969% and the highest interfacial area of 14.998 nm^−1^ were observed in the 30RFPC.

#### 3.3.2. Mechanical Properties of RFPCs

The effects of RF incorporation and ASO treatment on the mechanical properties of the RFPC were estimated. Figure 4 exhibits the tensile strengths (Figure 4a) and tensile modulus (Figure 4b) of the PP and RFPCs with varying ramie fiber fractions. It is observed that ASO modification and RF incorporation increase the tensile strength and tensile modulus of RFPCs. The tensile strengths of the modified 5RFPC, 10RFPC, 15RFPC, 20RFPC, 25RFPC and 30RFPC have increased by 3.4%, 6.6%, 7.3%, 9.0%, 11.1%, and 18.2%, respectively, compared to that of the unmodified 5RFPC, 10RFPC, 15RFPC, 20RFPC, 25RFPC and 30RFPC. The tensile strengths of the modified 5RFPC, 10RFPC, 15RFPC, 20RFPC, 25RFPC and 30RFPC increase by 10.9%, 18.8%, 23.0%, 27.2%, 37.2%, and 50.3% respectively, compared to that of PP. Because a good fiber/matrix bonding is generated from the ASO treatment making effective stress transfer from resin to fiber as the SEM results.

The flexural strength and flexural modulus of the neat PP and RFPCs are shown in Figure 4c,d. It is seen that ASO treatment and incorporation of the ramie fiber increases the flexural strength and flexural modulus of RFPCs. The flexural strengths of modified 5RFPC, 10RFPC, 15RFPC, 20RFPC, 25RFPC and 30RFPC are about 0.8%, 4.0%, 4.9%, 7.3%, 10.3%, and 11.9% higher compared to that of the unmodified 5RFPC, 10RFPC, 15RFPC, 20RFPC, 25RFPC and 30RFPC, respectively. The flexural strengths of the modified 5RFPC, 10RFPC, 15RFPC, 20RFPC, 25RFPC and 30RFPC increase by 1.9%, 8.8%, 12.6%, 17.2%, 22.9%, and 29.0% compared to that of PP, respectively. Flexural strengths of RFPCs improve with increment of RF addition caused by reinforcement effect and stiffer of fibers [38].

Figure 4e presents the elongation at break of the neat PP and RFPCs with different fiber fractions in tensile tests. The results show that the elongation at break drops dramatically with the addition of RF. The higher fiber addition, the lower improvement of the elongation at break. This is because the process of tensile deformation is essentially the process of consuming the flexibility of polymer chain [39]. Elongation at break of the RFPC is mostly determined by the flexibility of PP molecular chain and addition of RF hinders the flexibility of PP molecular chain [40]. It was also found that ASO treatment increases the elongation at break of RFPCs. Especially when fiber content is 5 wt.%, the elongation at break of modified 5RFPC is about 84.8% higher compared to that of the unmodified 5RFPC. The modification improved the fiber/matrix bonding at interface, which leaded to the effectively stress transfer from matrix to fiber and prevented the cracks propagation [41]. Therefore, the modified composites had a higher elongation.

The impact strengths of the neat PP and RFPCs with different fiber contents are presented in Figure 4f. Impact strength of RFPCs is lower compared to that of the neat PP due to lower impact strength of RF than PP, while the impact strength of the ASO modified RFPCs are higher than that of the unmodified RFPCs at the same fiber content. The impact strengths of modified 5RFPC, 10RFPC, 15RFPC, 20RFPC, 25RFPC and 30RFPC are about 10.9%, 13.6%, 16.4%, 23.7%, 26.2% and 32.7% higher compared to that of the unmodified 5RFPC, 10RFPC, 15RFPC, 20RFPC, 25RFPC and 30RFPC, respectively. Because modified RFPCs prevented crack expansion under the impact force due to better interlocking between the fiber and the matrix through ASO treatment [42].

An increase in RF fraction in RFPCs provides more opportunity for the interaction and stress transferring between fiber and matrix, which finally leads to the improvements of the tensile and flexural strengths in RFPCs. FTIR and SEM results also confirmed that the mechanical properties of ASO modified RFPCs are higher than those of unmodified RFPCs. Because ASO modification improves compatibility between fiber and matrix, which results in a better bonding quality at interface. Therefore, modification of RFs with the ASO emulsion strengthens the interfacial adhesion between treated RF and PP resin as discussed above, which further contributes to an efficient stress transferring in the RFPCs and results in higher mechanical properties.

### 3.4. Dynamic Mechanical Analysis (DMA)

#### 3.4.1. Storage Modulus (*E*′)

Storage modulus (*E*′) is typical of elastic response of materials and indicates the ability of the materials to maintain the energy [7]. It provides information about stiffness, rigidity and fiber/matrix adhesion of the composite material [43]. There are three regions in *E*′ while increasing the temperature. They are glassy region, glass transition region, and rubbery region. In the glassy region, *E*′ is high and composite is rigid due to the close-packed PP molecular chains. During the glass transition region, *E*′ is decreased dramatically around *T_g_* because of the PP polymeric chain movement. In rubbery region, *E*′ changes a little due to even more PP molecules movements at higher temperature [27]. Movement in PP molecular chain affects the viscoelastic properties of composites and adhesion between fiber and matrix.

Figure 5a represents *E*′ vs. temperature for the neat PP and ASO modified RFPCs at a fixed frequency of 1 Hz. *E*′ is rising with the increase of the RF loading revealing prominent reinforcement effects of RF. This is associated with the stiffness of RF. *E*′ goes up with the rise of fiber incorporation due to the mobility of the PP chains hindered by fiber fillers [30,43]. A decreasing trend is also observed in *E*′ of the fabricated RFPCs with the increase in temperature. According to properties of polymer, molecules become more active and the forces between them become weaker when temperature rises, which causes the decrease of fiber/matrix bonding as well as *E*′.

Mathematical prediction models of E’ of composite materials (rule of mixture, Einstein model, Guth model [44] and Kerner model [37]) are described by following Equations (5)–(8), respectively:(5)$$E^{\prime }=E_{m}^{\prime }\left(1+V_{f}\right)$$
(6)$$E^{\prime }=E_{m}^{\prime }\left(1+2.5V_{f}\right)$$
(7)$$E^{\prime }=E_{m}^{\prime }\left(1+1.25V_{f}+14.1V_{f}^{2}\right)$$
(8)$$E^{\prime }=E_{m}^{\prime }\left(1+V_{f}/\left(1-V_{f}\right)\right)\left(15\left(1-\gamma_{m}\right)/\left(8-10\gamma_{m}\right)\right)$$

Figure 6 displays *E*′ values of modified RFPCs using the equations mentioned above and compared to the value obtained from the dynamic mechanical analysis. It can be seen that experimental values are between the results obtained by Guth’s and Kerner’s models. Experimental data noticeably exceeded the values predicted by the rule of mixture, pointing to the interfacial adhesion enhanced by modification.

Using the storage modulus values, we can calculate different parameters like effectiveness coefficient C, degree of entanglement *ϕ* and reinforcing efficiency *r*. [45] Effectiveness coefficient *C* evaluates the effectiveness of fiber reinforcement. The higher value of *C* means the lower effectiveness of fiber dispersion in matrix. *C* can be calculated by Equation (9) [46]:
(9)$$C=\frac{{\left(E_{G}^{\prime }/E_{R}^{\prime }\right)}_{composite}}{{\left(E_{G}^{\prime }/E_{R}^{\prime }\right)}_{resin}}$$
where $E_{G}^{\prime }$
and $E_{R}^{\prime }$ are *E*′ in the glassy region and rubbery region, respectively. The temperature chosen for glassy region is −60 °C. Table 4 displays the *C* of modified RFPCs with different fiber content at different rubbery temperatures. The value of *C* presents a decrease trend with the increase of the temperature. This indicates that fibers impose an inferior dispersion effectiveness at a lower temperature. On the other hand, the value of *C* decreases with the rise of the fiber addition. The lowest *C* is obtained by the 30RFPC indicating a highest efficiency of fiber reinforcement. When the fiber addition is lower, the constraints of fiber throughout the matrix-rich regions are less efficient and result in higher *C* values. 

The entangled fiber network can immobilize the polymer chains so that giving rise to high degree of reinforcement. So, degree of entanglement *ϕ* of ramie fiber over the thermoplastic matrix is important to understand the fine dispersion of fibers in the matrix. The degree of entanglement *ϕ* can be calculated by Equation (10) [45]:
(10)$$\varphi =E_{G}^{\prime }/6RT$$
where $E_{G}^{\prime }$ is the *E’* in the rubbery region (100 °C), *R* is universal gas constant and *T* is temperature at Kelvin scale.

Reinforcing efficiency is to evaluate the reinforcing ability of fibers on the composite, which can be obtained using Einstein equation [47]:
(11)$$r=(E_{c}^{\prime }/E_{m}^{\prime }-1)/V_{f}$$
where $E_{c}^{\prime }$ and $E_{m}^{\prime }$ are values of *E’* of the composite and matrix in rubbery region (100 °C), respectively. The degree of entanglement *ϕ* and reinforcing efficiency *r* with different fiber loadings are shown in Table 5. The degree of entanglement becomes more effective with increase in the fiber loading indicating an effective dispersion in matrix with fiber loading. The reinforcing efficiency *r* reaches a maximum value of 15.295 at 10 wt.% of fiber content followed by a decrease at 15 wt% of fiber loading because of fiber agglomeration.

#### 3.4.2. Loss Modulus (*E*″)

*E*″ is an indication of the energy dissipation which is regarded as the viscous response of materials [46]. Figure 5b displays the *E*″ vs. temperature for the neat PP and RFPCs. The *E*″ increases with the rise of fiber fraction, because the presence of short ramie fibers creates fiber/matrix interfacial areas where energy is dissipated. A relaxation peak around 5 °C is observed for each loss modulus curves of the RFPCs. The higher *T_g_* and loss modulus with higher content of RF shows the modified RF improve the thermal properties and loss modulus of RFPCs in Table 6. The highest loss modulus and *T_g_* was observed for 30RFPC. The comparison study of PP with RFPCs in Table 6 indicates that incorporation of RFs exhibits an increase in loss modulus in the relaxation region. This is probably due to more energy dissipation at interfaces with an increase in internal friction through fiber incorporation [44].

#### 3.4.3. Damping Factor (*tan δ*)

The *tan δ* is the ratio of the loss modulus to the storage modulus during a dynamic loading cycle, which is a decisive parameter that presents the viscoelasticity and damping capacity [48]. The dependent variable *tan δ* vs. temperature for all composites is displayed in Figure 5c. In general, PP has three relaxation peaks (γ-relaxation peak, β-relaxation peak, and α-relaxation peak) in the curve of *tan δ* as temperature rises [48]. It is observed that pure PP and RFPCs show β-relaxation peaks (*T*_g_) and α-relaxation peak at about 15 °C and 80 °C, respectively. The *T*_g_ around 15 °C relates to motions of PP chain segments in the amorphous phase. The α-relaxation peak around 80 °C relates to the transition of the PP crystalline parts. For the PP and all the composites in the β-relaxation (glassy-rubbery transition region), the value of *tan δ* increases as the temperature increases until it reaches critical values and then decrease.

As seen in Figure 5c, incorporation of ramie fiber reduces the values of *tan δ* around *T*_g_ compared to the neat PP. Since RFs treated with the ASO, the compatibility between fiber and PP has been improved. The fibers limit the movement of the PP molecules and further result in the increase of *T*_g_ and a reduction of the value of *tan δ*. It can be concluded that a better interfacial bonding in composites indicates a lower value of damping factor.

It is also noticed that the higher the addition fraction of ramie fiber, the lower value of *tan δ* at *T*_g_ listed in Table 6. The increase of the fiber content in the modified RFPCs provides more opportunity for the interaction and stress transfer in interface. Furthermore, modified fibers also give more restriction on the movement of the PP matrix molecules. This suggests that the ASO treatment on the fibers can create a stronger bonding at interface and promote interactions between the fiber and matrix, which is confirmed by the results of SEM in Figure 3. Therefore, fiber addition leads to a better fiber/matrix bonding, a lower damping factor around *T*_g_ and a higher damping factor around α-transition temperature. On the other hand, RFPC’s viscoelastic characteristics increase internal friction and the additional viscoelastic energy dissipation, which may result in widen of transition region.

The *tan δ* peak in glassy transition region in Figure 5c results from the viscous movement of PP chains. The reduction of the *tan δ* peak and the broadening of peaks indicates a decrease of active PP chains and an improved interface bonding, hence can be used to estimate constrained chains. The volume fraction of constrained region *C_v_* can be obtained by Equation (12) [44]:
(12)$$C_{v}=1-\left(1-C_{0}\right)\frac{W}{W_{0}}$$
where *C_v_* and *W* are the volume fraction of the restricted mobility of polymer chains and the energy fraction loss in the composite, respectively. *C*_0_ and *W*_0_ are the volume fraction of the constrained region and the energy fraction loss and for pure PP, respectively. The energy loss fraction *W* can be calculated by Equation (13):(13)W=πtanδ/(πtanδ+1)

The values of constrained chain volume are increasing with fiber corporation in Table 6. Such an effect is caused by the hinder of fibers to restrict the polymer matrix. As shown in Figure 7, a fiber content and higher interface surface area leads to a higher *C_v_*. Because the ASO modification makes surfaces of the fiber rougher. The interlocking between matrix and fiber caused by rougher surfaces strengthened the interfacial adhesion.

The adhesion efficiency can be calculated by the adhesion factor *A*, which is shown in the following equation [37]:
(14)$$A=\frac{1}{1-V_{f}}\frac{tan\delta_{c}}{tan\delta_{p}}-1$$
where *tan δ_c_* and *tan δ_p_* are the relative damping ratio of the composite and the polymer obtained from the *tan δ* curves at the given temperature, respectively.

The lower value of *A* indicates a better interaction at the matrix-fiber interface. The negative value of *A* is due to anisotropy of fiber and improvement of interface region of composite. Table 6 shows the value of *A* decreases with the increase of the fiber loading. The 30RFPC has the lowest adhesion factor *A*, which means it has the better interaction at the matrix-fiber interface, which also contributes to a better stiffness and a higher storage modulus. 

#### 3.4.4. Cole-Cole Plot

Cole-Cole plot was plotted to understand the structural changes of composites after adding fibers, which revealed system homogeneity of composites [49,50]. Semi circles of the composites demonstrated a poor interfacial bonding between fiber/matrix, while imperfect semi circles display a heterogeneous system [51]. The nature of the composite system is represented by plotting *E*″ against *E*′ on the Cole-Cole plot in Figure 5d. Figure 5d shows the Cole-Cole plot of RFPCs with different ramie fiber loading. The curve of neat PP is closed to semi-circle. The higher fiber content, the curve is far away from semi-circle due to presence of fibers and interface. It indicates that RFPCs are heterogeneity systems and have more complicated viscoelastic characters with the incorporation of RFs. The incorporation of modified RFs into the composites significantly influences the shape of the Cole–Cole plot and improved the dynamic properties of RFPCs.

## 4. Conclusions

The impacts of modification and fiber addition on interface interaction, interface microstructure, mechanical properties, and viscoelastic properties of the RFPCs were studied in detail. The results demonstrated that ASO emulsion could decrease the amount of active hydroxyl groups and increase the roughness of ramie fiber and thus strengthening the interfacial adhesion on interface. The results showed a good interface bonding after fiber treatment with ASO, which contributed to an efficient stress transferring between matrix and fiber in the RFPCs and result in better mechanical properties. The ASO treatments and fiber addition increase the internal friction and generate additional energy dissipation, which enhanced the mechanical and dynamic mechanical properties compared to unmodified RFPCs. This kind of composite materials is expected to be used in vehicle industry to meet the demand of lightweight and sufficient performance at the same time.

## Figures and Tables

**Figure 1 polymers-13-04083-f001:**
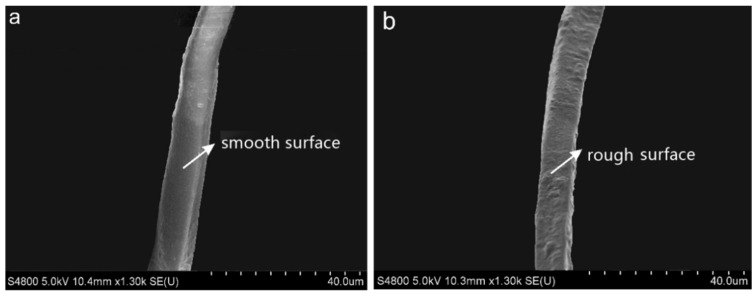
SEM images of the ASO emulsion treated and untreated ramie fibers: (**a**) untreated fiber, (**b**) treated fiber.

**Figure 2 polymers-13-04083-f002:**
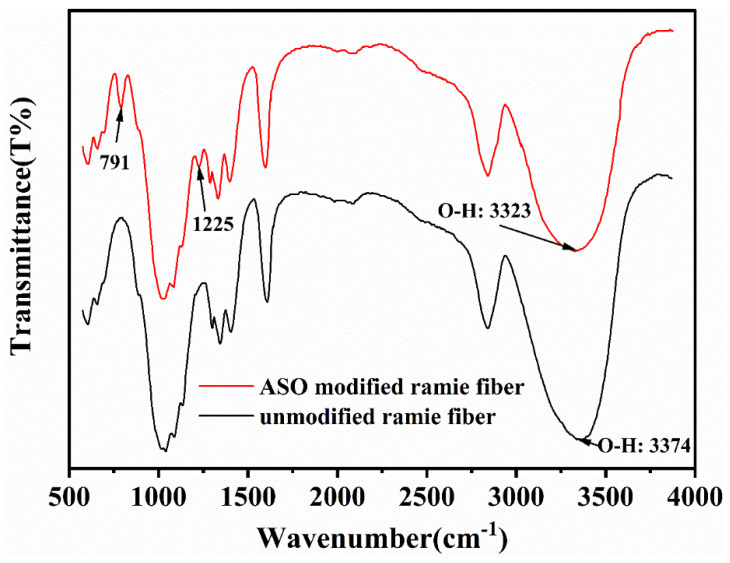
FTIR spectra of unmodified and ASO modified ramie fibers.

**Figure 3 polymers-13-04083-f003:**
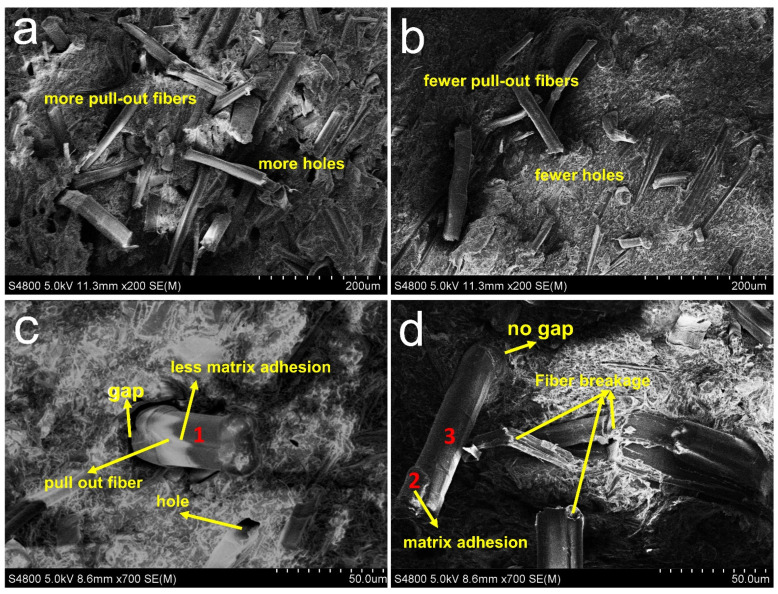
SEM of the fractured surfaces of modified RFPCs: (**a**,**c**) unmodified RFPCs, (**b**,**d**) ASO modified RFPCs.

**Figure 4 polymers-13-04083-f004:**
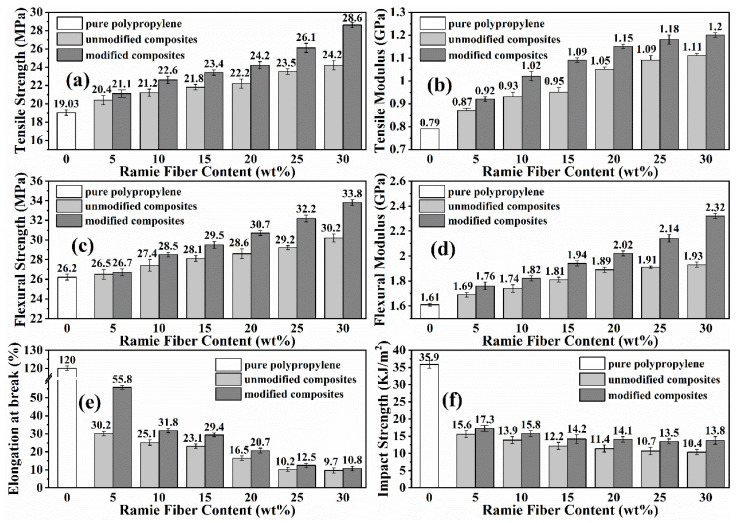
Mechanical properties of the neat PP, unmodified and modified RFPCs with different fiber fractions (**a**) tensile strength, (**b**) tensile modulus, (**c**) flexural strength, (**d**) flexural modulus, (**e**) elongation at break, (**f**) impact strength.

**Figure 5 polymers-13-04083-f005:**
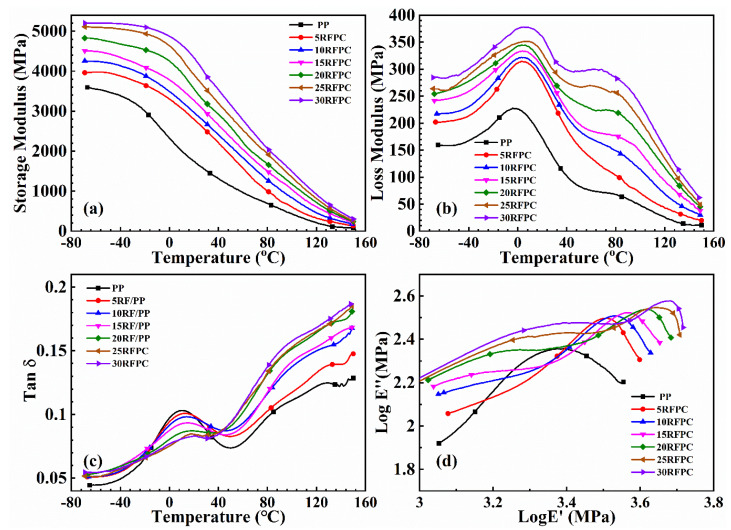
Dynamic mechanical properties of the neat PP and modified RFPCs with different fiber fractions (**a**) storage modulus, (**b**) loss modulus, (**c**) *tan δ*, (**d**) Cole-Cole plot.

**Figure 6 polymers-13-04083-f006:**
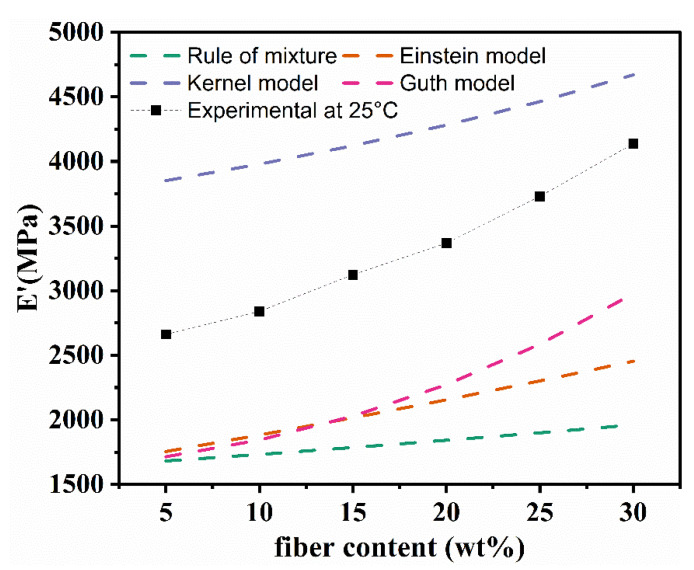
Experimental and predicted values of *E’* for modified RFPCs.

**Figure 7 polymers-13-04083-f007:**
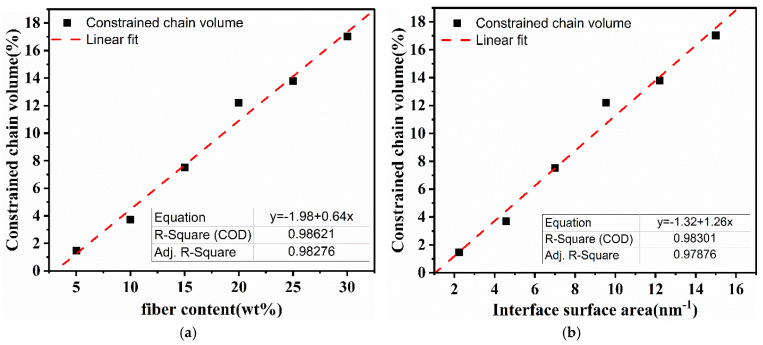
Constrained chain volume of prepared composites as a function of (**a**) fiber weight fraction, and (**b**) interface surface area.

**Table 1 polymers-13-04083-t001:** Properties of ramie fiber (RF).

Technical Specification	Value
Fiber length/mm	3–5
Mean diameter/μm	20–80
Density/g·cm^−3^	1.55
Young’s modulus/GPa	61.4–128
Tensile strength/MPa	400–1000

**Table 2 polymers-13-04083-t002:** Properties of polypropylene (PP).

Technical Specification	Value
Melt-flow index (MFI)/g·(10 min)^−1^	0.33
Melting point/°C	170
Density/g·cm^−3^	0.904
Poisson ratio	0.42
Young’s modulus/MPa	1550

**Table 3 polymers-13-04083-t003:** Surface elemental analysis of RFs from unmodified and modified RFPCs from EDX.

Sample	C (wt.%)	O (wt.%)	Si (wt.%)	O/C
unmodified RFPC (point 1)	57.5	46.5	-	0.81
modified RFPC (point 2)	76.5	17.9	5.6	0.23
Modified RFPC (point 3)	48.3	32.8	18.9	0.68

**Table 4 polymers-13-04083-t004:** Physical properties of neat PP and modified RFPCs.

Parameter	PP	5RFPC	10RFPC	15RFPC	20RFPC	25RFPC	30RFPC
Theoretical density, g/cm^3^	0.904	0.919	0.938	0.959	0.980	1.003	1.026
Experimental density, g/cm^3^	0.904	0.911	0.928	0.945	0.957	0.963	0.975
Porosity, %	0	0.845	1.102	1.438	2.364	3.944	4.969
Fiber volume fraction *V_f_*, vol%	0	2.978	6.086	9.332	12.725	16.277	19.997
Average particle distance, μm	-	128.020	83.926	62.157	48.194	38.096	30.264
Interface surface area, nm^−1^	0	2.234	4.564	6.999	9.544	12.207	14.998

**Table 5 polymers-13-04083-t005:** Effectiveness coefficient (C), degree of entanglement *ϕ* and reinforcing efficiency *r* for different modified RFPCs.

Paramerters	5RFPC	10RFPC	15RFPC	20RFPC	25RFPC	30RFPC
*C* (100 °C)	0.806	0.619	0.527	0.499	0.462	0.427
*C* (110 °C)	0.802	0.817	0.741	0.456	0.425	0.386
*C* (120 °C)	0.734	0.55	0.441	0.4	0.374	0.337
*C* (130 °C)	0.597	0.467	0.365	0.34	0.316	0.284
degree of entanglement *ϕ*	0.032	0.045	0.056	0.063	0.072	0.080
reinforcing efficiency *r*	13.099	15.295	15.029	13.485	12.969	12.162

**Table 6 polymers-13-04083-t006:** Dynamic mechanical properties of neat PP and modified RFPCs.

Composite	*T*_g_ Obtained from *E*″ Curves (°C)	*E*″ at *T*_g_ (MPa)	*T*_g_ Obtained from *tan δ* Curves (°C)	*tan δ* at *T*_g_	Adhesion Factor	Constrained Chain Volume, %
PP	−4	227.7	10	0.103	-	-
5RFPC	2.7	314.5	13.2	0.101	0.011	1.474
10RFPC	4.8	321.6	14.1	0.098	0.013	3.712
15RFPC	5.5	333.4	15.3	0.093	−0.004	7.514
20RFPC	6.1	344.5	17.8	0.087	−0.032	12.2
25RFPC	7.9	351.4	18.9	0.085	−0.014	13.793
30RFPC	8.9	378.0	21.1	0.081	−0.017	17.027

## Data Availability

The data presented in this study are available on request from the corresponding author.

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
