# Peer review of "Mechanical and Dynamic Mechanical Properties of the Amino Silicone Oil Emulsion Modified Ramie Fiber Reinforced Composites"

_polymers, 2021, doi:10.3390/polym13234083_

Round 1

Reviewer 1 Report

Was the density of RF from some datasheet or analyzed? Because the deviation is very high, should be more precise. Then Authors could determine the theoretical and experimental density of materials, porosity, volume franction and use these values to analyze more deeply the dynamic mechanical performance of composites. For more detailed procedure please see the exemplary paper:

Hejna, A. Poly(ε-Caprolactone)/Brewers’ Spent Grain Composites—The Impact of Filler Treatment on the Mechanical Performance. J. Compos. Sci. 20204, 167. https://doi.org/10.3390/jcs4040167

Were fibers washed after modification?

What was the actual amount of ASO that was bonded to the surface of fibers? It is very important for the proper analysis.

"The content of carbon in untreated RFPC is lower than that of ASO treated RFPC due to the adhesion of PP matrix" - please explain how the adhesion to matrix affects the chemical composition of fibers, because it is very interesting.

During description of the mechanical performance please refer more to other literature works.

Why C coefficient values were only presented for elevated temperatures? Why not for the room temperature at which composites would be used probably?

As mentioned above, I recommend to present deeper discussion on DMA results.

Author Response

Response to comment 1:

Q1.Was the density of RF from some data sheet or analyzed? Because the deviation is very high, should be more precise. Then Authors could determine the theoretical and experimental density of materials, porosity, volume fraction and use these values to analyze more deeply the dynamic mechanical performance of composites. For more detailed procedure please see the exemplary paper:

Hejna, A. Poly(ε-Caprolactone)/Brewers’ Spent Grain Composites—The Impact of Filler Treatment on the Mechanical Performance. J. Compos. Sci. 2020, 4, 167. https://doi.org/10.3390/jcs4040167

Response: Thanks very much for reviewer’s helpful suggestions. The density of RF was obtained from data sheet, we narrowed it to a precise value in the revision. As shown in page 3, Table 1. We have added the theoretical and experimental density of materials, porosity, volume fraction in Table 4 and used these values to analyze more deeply the dynamic mechanical performance of composites in the revised manuscript. As shown in page 7, section 3.3.1, Table 4; page 9-11, section 3.4.1; page 12, section 3.4.3, line 502-534.

Q2.Were fibers washed after modification?

Response: The fibers were not washed after modification. The modification procedure is presented in page 3, section 2.2.

Q3.What was the actual amount of ASO that was bonded to the surface of fibers? It is very important for the proper analysis.

Response: Thanks very much for your suggestions. The results of elemental analysis of unmodified and modified fiber in EDX gives a clear insight into the wt% of element Si bonded to the surface of fibers. We have added this information in Figure 3 and Table 3 in page 6 in the revision.

Q4."The content of carbon in untreated RFPC is lower than that of ASO treated RFPC due to the adhesion of PP matrix" - please explain how the adhesion to matrix affects the chemical composition of fibers, because it is very interesting.

Response: Thanks very much for reviewer’s helpful suggestions. We have marked the test points in Figure 3 so that we can see the matrix adhesion in our test point 2 in the Figure 3(d). Because the carbon content in PP is higher than that of fibers, the carbon content at point 1 of untreated fiber is lower than that at point 2 of treated fiber. We have given the explanation in the revision, as shown in page 6, section 3.2, Paragraph 1, line 209-213, Figure 3 and Table 3.

Q5.During description of the mechanical performance please refer more to other literature works.

Response: Thanks very much for your helpful suggestions. We have added literature works [38]-[42] when describing mechanical performances in section 3.3 “Mechanical properties of RFPCs” in the revision,as shown in page 8, section 3.3.2, line 276, 282, 284, 288 and 298; page 15, references section, line 613-618.

Q6. Why C coefficient values were only presented for elevated temperatures? Why not for the room temperature at which composites would be used probably?

Response: Thanks very much for reviewer’s careful review. According to the equation of C factor: , E’G is the storage modulus in glassy region and E’R is the storage modulus in rubbery region. The temperature in table 5 is the different temperatures chosen in the rubbery region for calculating C. The temperature chosen for glassy region is -60°C, which has been added in the revision in page 10, section 3.4.1, line 361-363. Using different rubbery temperatures of C values is to look at the effect of temperature on C factor. It can be seen from the Figure 5, 25 oC is in the transition state, so it did not be chosen for calculating C. In order to make this clear, we added the glassy temperature in Table 5 in the revision,as shown in page 11, Table 5.

Q7.As mentioned above, I recommend to present deeper discussion on DMA results.

Response: Thanks very much for reviewer’s helpful suggestions. We have added deeper discussions on DMA in the revision. We added the physical properties of RFPCs and used these values to analyze the porosity and interface surface area in composites of different fiber content in the revision, as shown in page 7, section 3.3.1, Table 4; We also compared the value of experimental date of E’ with values of four mathematical models and used the E’ to calculate effectiveness coefficient C, degree of entanglement Ï• and reinforcing efficiency r to analyze how fiber content effects properties of RPFCs in the revision, as shown in page 9-11, section 3.4.1 and Table 5; We used the values of tanδcalculating the adhesion factor and constrained chain volume to further obtain the interfacial properties of RFPCs, as shown in page 12-13, section 3.4.3, line 504-536.

Reviewer 2 Report

Research by He et al reports the mechanical and dynamic-mechanical properties of polypropylene matrix composites filled with different amounts of natural ramie fibers as such and pre-treated with amino silicone oil emulsion to improve the fiber-matrix interface in the products. The results obtained certainly provide a further interesting contribution towards the development of new natural fiber polymer composites of well-known interest in many industrial fields.

The results obtained with appropriate procedures are clearly reported and properly explained.

Despite the premises, I report minor revisions to be addressed befor epublication of this manuscript.

Lines 122-123: Please check the construction of the sentence. The verb is placed in a position that makes the text not readily understandable.

Paragraph 2.1.1 “Mechanical testing”: the authors mistakenly write the phrase "... unmodified and unmodified RFPCs ..." a couple of times instead of "... unmodified and modified RFPCs ...". Please correct.

Paragraph 3.3.2 “Cole-Cole plot” – Lines 342-344: in this first sentence the authors mention the phenomena of "... molecular cross-linking ..." which can be observed with this type of plot in the case of thermosetting composites. Since, apparently, the authors in this first sentence want to indicate the purpose for which the representation of the data according to Cole-Cole was taken into account, dealing with thermoplastic composites, it is suggested to limit the sentence to the understanding of the structural changes deriving from the inclusion of ramie fibres.

Author Response

Response to comment 2:

Research by He et al reports the mechanical and dynamic-mechanical properties of polypropylene matrix composites filled with different amounts of natural ramie fibers as such and pre-treated with amino silicone oil emulsion to improve the fiber-matrix interface in the products. The results obtained certainly provide a further interesting contribution towards the development of new natural fiber polymer composites of well-known interest in many industrial fields.

The results obtained with appropriate procedures are clearly reported and properly explained. Despite the premises, I report minor revisions to be addressed before publication of this manuscript:

Q1.Lines 122-123: Please check the construction of the sentence. The verb is placed in a position that makes the text not readily understandable.

Response: Thanks very much for your careful review. We have revised the sentence in line 122-123 and put the verb in a suitable place in the revised manuscript, as shown in page 3, section 2.2, line 122-123.

Q2.Paragraph 2.1.1 “Mechanical testing”: the authors mistakenly write the phrase "... unmodified and unmodified RFPCs ..." a couple of times instead of "... unmodified and modified RFPCs ...". Please correct.

Response: Thanks very much for your reminder. We have revised the mistakes and changed “unmodified and unmodified RFPCs” to “unmodified and modified RFPCs” in the revision, as shown in page 4, section 2.4.3, line 150 and line 154.

Q3.Paragraph 3.3.2 “Cole-Cole plot” – Lines 342-344: in this first sentence the authors mention the phenomena of "... molecular cross-linking ..." which can be observed with this type of plot in the case of thermosetting composites. Since, apparently, the authors in this first sentence want to indicate the purpose for which the representation of the data according to Cole-Cole was taken into account, dealing with thermoplastic composites, it is suggested to limit the sentence to the understanding of the structural changes deriving from the inclusion of ramie fibers.

Response: Thanks very much for your helpful suggestions. We have deleted "... molecular cross-linking ..." in the revised manuscript. See page 13, section 3.4.4, line 536.

Round 2

Reviewer 1 Report

Paper in order after corrections.